# Crohn’s Disease and Jejunal Artery Aneurysms: A Report of the First Case and a Review of the Literature

**DOI:** 10.3390/medicina58101344

**Published:** 2022-09-24

**Authors:** Paolo Vincenzi, Diletta Gaudenzi, Luca Mulazzani, Alberto Rebonato, Alberto Patriti

**Affiliations:** 1Department of Surgery, Ospedali Riuniti Marche Nord, Piazzale Cinelli n 1, 61121 Pesaro, Italy; 2Department of Perioperative Services, AOU Ospedali Riuniti di Ancona, Via Conca n 71, 60126 Ancona, Italy; 3Department of Radiology, Division of Interventional Radiology, Ospedali Riuniti Marche Nord, Piazzale Cinelli n 1, 61121 Pesaro, Italy

**Keywords:** visceral artery aneurysms, jejunal artery aneurysms, ileal artery aneurysms, visceral artery pseudoaneurysms, inflammatory bowel disease, Crohn’s disease, endovascular treatment, surgery

## Abstract

*Background and Objectives*: Jejunal artery (JA) and ileal artery (IA) aneurysms constitute less than 3% of all visceral artery aneurysms (VAAs), carrying a risk of rupture as high as 30%, and a mortality of 20%. Though many etiologies have been reported in the literature, no mention exists on a causal association between these aneurysms and inflammatory bowel diseases (IBD). We present the first case of a JA aneurysm related to Crohn’s Disease (CD) together with a review of the literature. *Materials and Methods*: A 74-year-old male presenting with CD intestinal relapse and an incidental finding at the computed tomography enterography (CTE) of a 53 × 47 × 25mm apparently intact JA pseudoaneurysm, arising from the first and second jejunal branches, underwent coil embolization followed by small bowel resection, with an uneventful outcome. We also included the review of literature on JA and IA aneurysms, analyzing all reports published in PubMed and Scopus from 1943 to July 2022. *Results*: 60 manuscripts with 103 cases of JA and IA aneurysms in 100 patients were identified. Among cases with available data, 34 (33.0%) presented acutely with rupture, 45 (43.7%) were described as non-ruptured. 83 (80.6%), and 14 (13.6%) were JA and IA aneurysms, respectively, having a median size of 15 (range:3.5–52) mm. Atherosclerosis (16.5%), infections (10.7%), and vasculitides/connective tissue disorders (9.7%) represented the main causes mentioned. Mean age was 53.6 (±19.2) years, male patients being 59.4%. One third of patients (32.4%) were asymptomatic. Overall, treatment was indicated in 63% of patients, with surgery and endovascular procedures performed in 61.9% and 38.1% cases, respectively. The technical success rate of endovascular treatment (EVT) was 95.8%. The mortality rate was 11.8%, being higher (21.2%) in the rupture group. *Conclusions*: The prompt treatment accomplished in our case granted a successful outcome. JA and IA aneurysms should be included among local complications of IBD. Considering their high potential for rupture, regardless of size, a low threshold for endovascular or surgical treatment should be applied.

## 1. Introduction

The prevalence of visceral artery aneurysms (VAAs), i.e., a dilatation to the extent of at least 1.5 times the size of the original vessel involving the main trunk of splanchnic arteries and/or their branches [1], is estimated to be low, ranging between 0.01 and 2% on autopsy series [2,3], though their incidental finding, particularly in asymptomatic patients, have been increasingly reported, in relation to a widespread application of the modern and sophisticated radiologic imaging [2].

Distribution of VAAs varies greatly among the different series, with splenic artery (SA) aneurysms being the most common (30–60%), followed by celiac trunk (CTA) (2–46%), hepatic (HA) (4–30%), superior mesenteric (SMA) (2–9%), and inferior mesenteric artery (IMA) (1%) aneurysms [1,2,4,5,6].

Among branch aneurysms, those arising at the level of pancreaticoduodenal (PDA) and gastroduodenal (GDA) arteries are the most frequently detected, with a prevalence varying between 2 and 15%, whereas those involving the gastric-gastroepiploic (GA-GEA), jejunal (JA), ileal (IA), and colic arteries (CA) have the lowest incidence, documented at 2–5% [1,2,4,5,6].

Aneurysm multiplicity, defined as a concomitant presence of other aneurysms in a separate vascular bed [2], might occur in approximately 20–50% of all VAAs [1,2,7].

Histopathologically, aneurysms are broadly divided into true and false. The first, by locally expanding while maintaining intact all the three components of the arterial wall, are primarily caused by atherosclerosis and fibromuscular dysplasia [2].

Conversely, pseudoaneurysms are unstable growths within the adventitial layer secondary to a damage of the intimal layer, following trauma, infection, vasculitis, inflammation, or iatrogenic causes [2].

While the current routinary adoption of procedures involving the liver, biliary tree, and pancreas have led to a significantly increasing documented incidence of iatrogenic pseudoaneurysms, especially at the level of HAs, PDAs, and GDAs [8,9], differently from infectious (mycotic) aneurysms, whose frequency is constantly reported decreasing [1], no mention exists in literature of a causal association between splanchnic aneurysms and inflammatory bowel diseases (IBD).

Indeed, despite the rising prevalence of IBD, especially Crohn’s disease (CD) and the accompanying transmural inflammation with penetrating disease to adjacent structures, leading to fistulas, strictures and abscesses, VAAs are not listed among the local complications of IBD [10,11,12].

Therefore, we present a rare case of a giant jejunal branch pseudoaneurysm associated with CD relapse. A complete and extensive review of all cases of JA and IA aneurysms cited in the English scientific literature is displayed as well.

## 2. Case Report

A 74-year-old male patient presented to our Emergency Department with 5-days history of gastrointestinal bleeding in the form of hematochezia and mild diffuse abdominal pain, not associated with positional changes.

At physical examination, he was alert, afebrile, with blood pressure of 125/70 mmHg, and heart rate of 65 beats per minute. His abdomen was soft, with moderate diffuse tenderness, in absence of rebound or guarding and palpable masses. Initial laboratory tests revealed leucocyte count of 10.6 × 10^3^/mmc, CRP of 0.7 mg/dL, hemoglobin of 7.7 g/dL, hematocrit of 22.6%, mean corpuscular hemoglobin (MCH) of 28.1, and red blood cells of 2.75 × 10^6^/mmc, for which he was transfused two units of packed red blood cells (PRBC) and started on injectable Vitamin K (Konakion^®^, Roche Pharmaceuticals, Basel, Switzerland) at a dose of 10 mg daily.

Since his past medical history was remarkable exclusively for long-standing mild ileal CD managed with Mesalazine (Pentasa^®^, Ferring Pharmaceuticals, Milan, Italy) at a dose of 1 gr three times daily, without significant disease activity, relapse episodes or complications, the patient was admitted to the Department of Gastroenterology and started on oral Budenoside (Intesticort^®^, Sofar Pharmaceuticals, Milan, Italy) at a dose of 9 mg daily.

Although hemodynamically stable, the patient had three further episodes of hematochezia, with hemoglobin level dropping to 7.4 g/dL, for which a third unit of PRBC was transfused.

In order to assess the disease extent, the patient underwent computed tomography enterography (CTE) showing a JA pseudoaneurysm of 53 × 47 × 25 mm, rising from the first and second jejunal branch of SMA. Neither signs of rupture nor free fluid were shown. No concomitant VAAs were described. Other findings included an active disease involving the jejunum, ileum, and the descending colon (Figure 1).

Based on such findings, the patient was transferred to the Department of General Surgery and the decision was made to perform urgent mesenteric angiography. The right common femoral artery was punctured with an 18-gauge needle using the standard Seldinger technique. The needle was withdrawn and a 5-Fr, 11 cm-long sheath was advanced into the artery. A 4-Fr. C2 catheter cobra type TEMPO^®^ (Cordis, Baar, Switzerland) was advanced to the SMA with over-the-guide-wire technique. Angiography clearly revealed the pseudoaneurysm arising from the proximal and distal branches of the first JA and from the second JA (Figure 2).

In the same session, the second JA was selectively catheterized by using a coaxial technique and through a 2.4-Fr microcatheter Progreat^®^ (Terumo, Tokyo, Japan), embolization was performed by using four Concerto^®^ (Medtronic, Minneapolis, MN, USA) complex helical microcoils (1 coil 2 mm/8 cm and 3 coils 3 mm/4 cm). An analogous procedure was repeated for the distal branch of the first JA (Figure 3).

Multiple attempts to catheterize the proximal branch as the only residual afferent to the aneurysm were unsuccessful, whereas embolizing the main trunk of the first JA would have led to a significant risk for infarction of the supplied intestine.

Hence, subsequent digital subtraction images demonstrated residual filling of the false aneurysm. 

Due to partial isolation of the aneurysm, we decided to perform urgent surgery. A middle laparotomy was carried out. The pseudoaneurysm was identified inside the mesentery of the jejunum. No hemoperitoneum was detected. The surrounding jejunum together with a portion of ileum for an extent of approximately 130 cm were affected by a severe transmural inflammation compatible with CD relapse, whereas the colon was normal. Based on such findings, we decided to resect this segment of small intestine.

Histologic examination confirmed the jejunal pseudoaneurysm characterized by disruption of the arterial wall and marked accumulation of endothelial and smooth muscle cells without signs of arteritis and/or necrotizing vasculitis. An adjacent mesenteric hematoma and a severe transmural inflammation compatible with CD were the associated histologic findings in the resected intestine.

Postoperative course was characterized only by mild anemization for which two units of PBRC were transfused and the patient was discharged on postoperative day five. 

Currently, at 3 months after surgery, the patient is doing well and does not present any recurrence of CD.

## 3. Materials and Methods

### 3.1. Literature Review

We performed a PubMed (National Library of Medicine, Bethesda, MD, USA) and Scopus (Elsevier, Amsterdam, The Netherlands) database search using the terms ‘‘jejunal artery aneurysm/pseudoaneurysm’’, ‘‘ileal artery aneurysm/pseudoaneurysm’’, “superior mesenteric artery aneurysm/pseudoaneurysm”, “visceral artery aneurysm/pseudoaneurysm” and “splanchnic artery aneurysm/pseudoaneurysm”, retrieving all articles published from 1943 to July 2022. Animal studies were excluded. 

We provided a flow chart outlining the process of identification of eligible case studies (Figure 4).

Data extracted were number of cases reported, patient demographics, aneurysm location (jejunal vs. ileal), type (true vs. false), size, etiology, clinical presentation, presence of concomitant aneurysms, management and treatment, mortality including aneurysm-related mortality.

### 3.2. Statistical Analysis

Frequency distributions were determined for baseline categorical variables, and the mean along with standard error (±SE) were calculated for baseline continuous variables.

## 4. Results

### 4.1. Study Population

The above-mentioned key words yielded a total of 103 cases in 100 patients from 60 manuscripts [4,6,13,14,15,16,17,18,19,20,21,22,23,24,25,26,27,28,29,30,31,32,33,34,35,36,37,38,39,40,41,42,43,44,45,46,47,48,49,50,51,52,53,54,55,56,57,58,59,60,61,62,63,64,65,66,67,68,69,70]. Most of the case studies (*n* = 52, 86.7%) were isolated case reports, with only 8 (13.3%) reporting at least 2 cases of jejunal and ileal branch aneurysms, generally within the context of larger series on splanchnic aneurysms, as outlined in Table 1.

Among the cases identified, rupture occurred in 34 (33.0%), whereas 45 (43.7%) cases were reported as not complicated by aneurysmal rupture, as shown in Table 1, Table 2 and Table 3. In the remaining 24 (23.3%), no information on the matter was available.

A total of 83 (80.6%) and 14 (13.6%) were JA and IA aneurysms, respectively, whereas the aneurysm site was not specified in the remaining 6 (5.8%).

Overall, among the 69 (69.0%) patients with documented demographics, mean age was 53.6 (±19.2) years, with a percentage of male patients being 59.4% (41/69).

Atherosclerosis (*n* = 17, 16.5%), infections (*n* = 11, 10.7%), vasculitides and connective tissue disorders (*n* = 10, 9.7%) represented the most frequent reported causes of aneurysm formation, followed by a congenital origin (*n* = 7, 6.8%), segmental arterial mediolysis (*n* = 5, 4.8%), and acute (*n* = 2, 1.9%) or chronic pancreatitis (*n* = 4, 3.9%).

Miscellaneous conditions were the other etiologies specified in 6.8% (*n* = 7) of cases, whereas in the remaining 38.9% (*n* = 40), the etiology was not mentioned or classified idiopathic, or it was not possible to determine from the patient characteristics.

True and false aneurysms were described in 30 (29.1%) and 34 (33.0%) cases, respectively, while no mention on the pathophysiology of the arterial dilatation was made in the other 39 (37.9%) cases.

Data regarding aneurysms dimension was available in 49 (47.6%) cases with a median size of 15 (range: 3.5–52) mm.

False aneurysms were slightly larger than true, having a median size of 15 (range: 4.8–50) mm compared to 13.5 (range: 3.5–52) mm.

In 16 of 76 patients (21.0%), a concomitant aneurysm was described in a separate vascular bed, including 11 (68.7%) with a complementary splanchnic aneurysm, 4 (25.0%) with an abdominal aortic aneurysm, and 1 (6.3%) with a renal, intracranial, lower extremity, intercostal and thoracic artery aneurysm, respectively.

Treatment, clinical picture, and outcomes of patients with multiple aneurysms are shown in Table 4.

Of the 74 patients (74.0%) with a disclosed clinical picture at diagnosis, 24 (32.4%) were asymptomatic, being the abdominal imaging performed for other reasons.

Among the symptomatic cases (*n* = 50, 67.6%), 23 (46.0%) and 6 patients (12.0%) experienced acute and chronic abdominal pain, respectively. Other common symptoms included gastrointestinal bleeding in the form of melena and/or hematochezia (*n* = 22, 44.0%) and hypotension/shock (*n* = 14, 28.0%), while fever and small bowel obstruction were documented in only two (4.0%) and one patient (2.0%), respectively.

At least two concurrent symptoms were mentioned in 15 patients (30.0%).

In 63.0% of patients (63/100), treatment of the jejunal and/or ileal artery aneurysm was indicated, including surgery in 39 patients (61.9%) and endovascular procedures in 24 patients (38.1%).

Technical success rate of EVT defined as immediate complete exclusion of the aneurysmal sac without the need to perform other procedures [71], regardless of endovascular or surgical, was reported in 95.8% of patients (23/24), as displayed in Table 5.

In the only patient with unsuccessful endovascular procedure, surgery was carried out.

The median size of treated aneurysms was 15 (range: 4–52) mm.

Among the 13 patients (13.0%) included in the observation cohort, data on natural evolution of aneurysms were presented in only 1 case (7.7%), with the true JA aneurysm remaining dimensionally stable at 12 months surveillance imaging.

Overall, death was reported in 9 patients (9/76, 11.8%). The leading cause for patient mortality was rupture of the aneurysm, jejunal or ileal as well as of a concomitant splanchnic artery (*n* = 5, 55.6%), followed by myocardial infarction (*n* = 2, 22.2%), recurrent episodes of pulmonary hemorrhage secondary to Microscopic Polyangiitis (*n* = 1, 11.1%) and an unknown reason (*n* = 1, 11.1%).

### 4.2. Rupture Group

Aneurysmal rupture was the acute clinical picture described in 34 cases (33.0%) involving 33 patients (33.0%). Gastrointestinal bleeding (16/33, 48.5%), acute abdominal pain (15/33, 45.4%), and hypotension/shock (13/33, 39.4%) represented the most common symptoms reported, associated with radiologic or operative detailed findings of hemoperitoneum and/or mesenteric hematoma in 11 patients (11/33, 33.3%).

A combination of at least two different symptoms was mentioned in approximately one third of patients (12/33, 36.4%), while the clinical picture at diagnosis was not available in only two patients (6.0%).

A total of 25 (73.5%) and 7 (20.6%) cases were JA and IA aneurysms, while in the other 2 (5.9%) the site was not specified.

In 30 of these 34 patients (88.2%), demographics were reported, with a practically equal distribution of gender, being male patients 16 of 30 (53.3%) and with a mean age of 48.6 (±19.1) years.

Main etiologies included: vasculitides and infections, each in 6 cases (17.6%), a congenital origin in 4 (11.8%), atherosclerosis and segmental arterial mediolysis, each in 3 (8.8%), and acute pancreatitis in 2 (5.9%).

True and false aneurysms were 10 (29.4%) and 17 (50.0%), respectively, while in the remaining 7 (20.6%) no mention was made on the pathophysiology of aneurysm formation.

Though dimensions were not reported in almost half of cases described (15/34, 44.1%), median size of ruptured aneurysms was 12 (range: 4–52) mm.

Among the 33 patients included, 5 (15.1%) presented at least 1 concomitant aneurysm in a separate vascular bed of whom all (100.0%) at least 1 associated VA and 1 (20.0%) concurrently a renal, cerebral, and thoracic artery aneurysm (Table 4).

Urgent treatment was performed in all patients, with 22 (66.7%) and 11 (33.3%) undergoing surgical repair and endovascular treatment (EVT), respectively.

Surgical treatment consisted of surgical ligation, aneurysmectomy, and small bowel or colonic resection in the majority of cases, whereas EVT was mainly represented by embolization with coils, reporting a technical success rate in all cases (100.0%).

Data on ruptured aneurysms are presented in Table 1 and Table 2.

The mortality rate was 21.2% (7/33), being directly related to the ruptured aneurysm in more than half of patients (*n* = 4, 57.1%), as shown in Table 5.

### 4.3. Non-Rupture Group

A total of 45 (43.7%) non-ruptured aneurysms in 43 patients (43.0%) were identified in our literature search.

The clinical picture at diagnosis included: 55.8% (24/43) being asymptomatic, 18.6% (8/43) reporting acute abdominal pain and 14.0% (6/43) presenting with chronic abdominal pain or at least one episode of gastrointestinal bleeding, respectively.

At least two symptoms were listed in three patients (3/19, 15.8%).

In total, 34 (75.6%) and 7 (15.6%) cases were JA and IA aneurysms, while the location of intestinal aneurysm was not available in the others 4 (8.8%).

In this cohort, of the 39 patients (90.7%) with mentioned demographics, 64.1% (25/39) were male, and mean age was 58.4 (±18.3) years.

Main causes were represented by atherosclerosis (14/45, 31.1%), followed by infections (5/45, 11.1%) and chronic pancreatitis and vasculitides/connective tissue disorders, each reported in 4 cases (8.9%).

A congenital origin (3/45, 6.7%), segmental arterial mediolysis (2/45, 4.4%) and miscellaneous conditions (4/45, 8.9%) were the other etiologies listed, while in the remaining 9 patients (20.0%), the underlying ethiopathogenesis was not available.

True and false aneurysms were 20 (44.4%) and 17 (37.8%), respectively, while no mention was made on the pathophysiology of aneurysm formation in the remaining 8 (17.8%).

Though dimensions were not reported in one of three cases (15/45), median size of non-ruptured aneurysms was 15 (range: 3.5–50) mm.

Concomitant aneurysms in a separate vascular bed were depicted in approximately one quarter of patients (11/43): more than half (6/11, 54.5%) having at least one associated VA, 4 (36.3%) having an abdominal aortic aneurysm and 1 (9.0%) having a coronary, cerebral and intercostal artery aneurysm, respectively, as displayed in Table 4.

A total of 30 patients (69.8%) had undergone elective operative repair of the aneurysm at a median size of 15 (range: 5–50) mm of whom 17 (56.7%) with open surgery and 13 (43.3%) with endovascular procedures.

The technical success rate of EVT was 92.3% (Table 5).

The mortality rate was 4.7% (2/43), the causes of death being MI in one patient and rupture of a concomitant PDA aneurysm in the second patient (Table 5).

Data on non-ruptured aneurysms are presented in Table 1 and Table 3.

## 5. Discussion

As far as we are aware, this is the first report of a visceral pseudoaneurysm directly related to CD. The prompt treatment accomplished, endovascular first by embolization and then surgical, might explain the successful outcome in our patient.

Indeed, though the overall incidence of splanchnic aneurysms has been increasing in recent years, particularly with regard to iatrogenic and incidentally detected categories [1,2,72], intestinal aneurysms have been constantly reported to be the most uncommon among VAAs, representing less than 3% of all cases [2,4,21,72], most of them identified and described in isolated reports.

Accordingly, in our literature search, we were able to identify only 103 cases of JA and IA aneurysms in 100 patients during an interval period of almost 80 years, with a definite predominance of the first type [4,6,13,14,15,16,17,18,19,20,21,22,23,24,25,26,27,28,29,30,31,32,33,34,35,36,37,38,39,40,41,42,43,44,45,46,47,48,49,50,51,52,53,54,55,56,57,58,59,60,61,62,63,64,65,66,67,68,69,70].

Even though Stanley et al. [4] reported those aneurysms being not gender-related and more common in elderly individuals, patients of male gender and with an average age of roughly 53 years were those more frequently affected in our analysis, being slightly older in the no-rupture group and tendentially younger without gender prevalence in the other group.

Indeed, though previously being generally described as asymptomatic [1,72,73], our review identified only one third of patients being completely asymptomatic while the other two third presented mainly with abdominal pain, hypotension/shock, and gastrointestinal bleeding.

Clearly, a non-complicated aneurysm was generally asymptomatic, with only one in five patients complaining of pain and gastrointestinal bleeding, whereas the same symptoms together with hypotension/shock represented the classic symptomatic triad accompanying a ruptured aneurysm.

Moreover, as stated by recent reviews [1,2,7,72], we confirmed the main etiologies of jejunal and ileal aneurysms being represented by atherosclerosis, a congenital origin, connective tissue disorders, vasculitides, and segmental arterial mediolysis, while infections, i.e., gastrointestinal tuberculosis and endocarditis, acted as significant determinants particularly in early reports.

If atherosclerosis emerged as the main cause in the non-rupture group, inflammatory conditions, promoting the development of pseudoaneurysms carrying a higher risk of rupture [1,2], were dominant in the group of complicated aneurysms.

Nevertheless, we could find in the literature any associations reported between IBD and intestinal aneurysms, as well as these vascular lesions have not been listed among the local complications of CD or UC, such as fistulas, abscesses, strictures, and colon cancer [10,11,12].

On the other hand, a recent review described an association, though uncommon, between vasculitides and IBD, reporting 32 cases identified through North America vasculitis databases search and 306 cases through PubMed literature search over the past 50 years [74].

Primary large-vessel vasculitides, especially Takayasu arteritis, represented the main group listed, followed by cutaneous and ANCA-associated vasculitis, being generally diagnosed several years after the IBD and presenting with constitutional and/or vascular symptoms, principally in women [74].

In our case, based on the proximity of the vascular lesion to the intestine involved by CD relapse and its absence at previous recent radiologic imaging, we assumed that the acute inflammation, extending to the adjacent mesentery, could involve the intestinal branches of SMA, leading to pseudoaneurysms development, though exclusively theoretical due to paucity of data published.

Indeed, the JA aneurysm (JAA) described in our case most likely represented an incidental finding at the CTE performed to assess the CD severity index, being the clinical picture described of hematochezia and acute abdominal pain principally related to the relapsing intestinal inflammatory disorder rather than the pseudoaneurysm itself.

However, though the SMA arteriography was not able to detect any contrast extravasation at the level of the JAA, the histologic examination showed a mesenteric hematoma, concerning for an initial rupture of the pseudoaneurysm, with potential higher rate of morbidity and mortality for our patient in absence of a definitive and rapid treatment.

Indeed, even if it has been difficult to establish the natural history of these vascular lesions due to the above-mentioned reasons, rupture might occur in up to one third of JA and IA aneurysms [33,73], with a reported mortality of 20% [2,4], in most cases directly related to the event of rupture, as confirmed in our literature review.

Furthermore, the relationship between aneurysmal size and rupture remains controversial, with historical data suggesting a high risk of rupture, strictly correlated to aneurysm dimensions [4,5,21] and, conversely, recent data indicating a more indolent course with a lower risk of rupture [6,7,70].

In particular, Pitcher et al. [70], in their extensive review of 144 true aneurysms involving the SMA and its branches, reported only two ruptures, of whom one interesting a small (12 mm) inferior PDA aneurysm, while no ruptures occurred in the group of 91 aneurysms <20 mm and in the group of twelve patients with aneurysms ≥20 mm managed conservatively and observed during a median follow-up of 73 months.

In addition, the Authors identified only 16 aneurysms with a rapid growth rate, defined as ≥2 mm/year, advocating to electively treat exclusively those true SMA aneurysms that are symptomatic, present a size ≥20 mm or a rapid growth rate [70], in contrast with the Society for Vascular Surgery clinical practice guidelines that recommend treating every type of SMA aneurysm, regardless of size, approach justified by a potential high risk of rupture ranging between 38 and 50%, according to the Authors [73].

Thus, the size criterion might not be sufficient to predict the risk of rupture, being other factors such as location, type, and etiology of the aneurysm crucial in determining this evolution [1,2].

Indeed, as emerged in our review, though with the important limitation of missing data, ruptured aneurysms were smaller than non-ruptured, with a higher number of pseudoaneurysms listed in the first group as well as more lesions caused by inflammatory and infectious diseases compared to true degenerative aneurysms documented mainly in the second group.

These findings are in agreement with Pitton et al. [8] and Shukla et al. [75] that reported a significantly higher risk of rupture for false VAAs compared to true VAAs (76.3% vs. 3.1% in Pitton et al., 81.8% vs. 35.3% in Shukla et al.) and with Pitcher et al. [70] whose very low incidence of aneurysmal rupture (1.4%) might be explained by excluding pseudoaneurysms in their review.

Strictly related to this point, the above-mentioned Society for Vascular Surgery clinical practice guidelines strongly advise to treat every visceral pseudoaneurysm, independently from the dimension or location, due to an intrinsically significant major risk of rupture [73].

Not surprisingly, in the largest series of ruptured splanchnic artery aneurysms to date, Shukla et al. [75] documented a significant difference in 30-day morbidity (1% vs. 13.7%; *p* = 0.003) and mortality (0% vs. 13%; *p* = 0.001) after repair of intact vs. ruptured VAAs.

Though with necessarily inferior numbers included in our literature review in relation to the area of research, we were able to confirm a higher mortality rate when any type of treatment was applied to ruptured JA or IA aneurysms compared to intact cases (21.2% vs. 4.7%) (Table 5).

In the main case series on VAAs, additional aneurysms were described in different locations in a range of 20–50% [1,2,7], including other splanchnic arteries as well as intracranial, lower extremities, coronary arteries, and abdominal aorta.

Similar conclusions were reached in our literature review with one in four patients presenting a concomitant aneurysm of whom approximately 70% with an associated splanchnic aneurysm and 40% with at least two additional aneurysms in different locations, stressing the importance of a thorough screening in all patients with VAAs to rule out concomitant lesions (Table 4).

Various therapeutic strategies are currently available for JA and IA aneurysms, including surgery, generally performed in an open fashion (aneurysmectomy, ligation, bypass with a venous graft or prosthetic vascular graft and intestinal resection), EVT (embolization and/or stent placement), or a combination of treatments (hybrid approach), depending on the anatomical characteristics such as size, site, features, and etiology of the aneurysm, as well as the patient’s presentation, comorbidities, and risk factors [1,2,72].

In recent years, endovascular approaches have been increasingly performed as first-line treatment, with technical success rates superior to 80%, low morbidity, and almost zero mortality, resulting in faster post-procedure recovery and consequently shorter hospital stays compared to open surgery [1,2,8,53,72].

Essential requisite of a successful EVT relies on completely blocking all inflow and outflow arteries, leading to the isolation of the aneurysm and thus to its occlusion [1,53].

Particularly, in the urgent setting of rupture and gastrointestinal bleeding, EVT has been demonstrated to be very useful, with the main advantages of being minimally invasive and repeatable, in case the first attempt might not be completely successful, of offering a precise localization of the aneurysm and of adequately assessing the collateral flow, whereas open surgery is associated with higher morbidity and mortality due to a significant technical complexity in this setting [53,72,74].

Reported complications and limitations of EVT include technical failure to catheterize the artery, high re-intervention rates, arterial thrombosis or embolism resulting in intestinal necrosis and perforation or late stricture requiring urgent operation or re-operation, coil migration and aneurysm recurrence, in association with a shortage of facilities with adequate resources for emergency treatments [1,2].

On the other side, open surgical repair of splanchnic aneurysms is a safe and durable treatment option, particularly in an elective setting, allowing for a real-time assessment of the distal flow to the organ [2].

Nevertheless, with advances in and ever-more expanding indications for EVT, surgery tends to be applied only when EVT is too difficult to perform [1].

In our review, in case of rupture, a surgical treatment was employed in approximately 70% of cases with the remaining patients undergoing an EVT, while in the elective context, surgery and EVT were almost equally distributed.

The reported technical success rate of the EVT was 100% in presence of ruptured aneurysms, whereas in an elective context, it was slightly inferior (92%), with surgery performed in only one patient after a single unsuccessful endovascular procedure (Table 5).

From what emerged in our review, intestinal aneurysms, regardless of clinical presentation, were generally managed with surgery in older reports, while recent reports have relied largely on endovascular procedures, both in an urgent and elective context, as expected (results not shown).

In our specific case, since coil embolization was only partially successful in isolating the JAA, we decided to proceed with surgery afterwards, in agreement with the current clinical practice guidelines of the Society for Vascular Surgery that strongly recommend, for JA and IA aneurysms, elective intervention when size is at least 20 mm in maximal diameter and emergent treatment, regardless of dimension, when symptomatic or ruptured and in presence of pseudoaneurysms [73].

Indeed, though a second endovascular approach was considered as a potential option alternative to surgery, the high risk of intestinal ischemia likely related to a complete occlusion of the first inflow artery, prevented us from repeating the endovascular procedure.

A different strategy would have been applied in case of a complete aneurysmal isolation during the first procedure, allowing avoidance of surgery, to preserve the intestinal integrity and to medically treat the CD relapse, as stated by the current guidelines on the management of IBD [10,11].

The above-mentioned clinical practice guidelines of the Society for Vascular Surgery stemmed from a systematic review and meta-analysis on all studies including at least 10 patients with VAAs conducted between 1980 and 2017, aiming at reporting the outcomes after open and/or endovascular treatment of splanchnic aneurysms [73].

The Authors identified only 15 cases of JA, IA, and CA aneurysms of whom 8 and 7 were managed by an endovascular and surgical approach, respectively [73], and since we included isolated case reports other than larger case series in a significantly broader time period, our literature review was able to report 103 cases of JA and IA aneurysms of whom 63 were treated by surgery or EVT.

Previous reviews on this theme were conducted by Stanley et al. [4] in 1970 documenting 17 cases of JA, IA, and CA aneurysms, including their 4 cases and by McNamara et al. [21] in 1980, describing 21 cases of JA and IA aneurysms, including those previously listed by Stanley et al., while, to the best of our knowledge, the last one, performed by Asano et al. [46] in 2008, reported exclusively 19 cases.

Therefore, according to our exhaustive and detailed analysis, though with the significant drawback of missing data, we might propose to electively treat JA and IA aneurysms even smaller than the threshold of 20 mm suggested by the current guidelines [73] since it appears that rupture occurs independently from the size of these very rare VAAs.

Likewise, other Authors made different recommendations and suggestions on the management of VAAs from those formulated by the Society for Vascular Surgery, among all Pitcher et al. [70] with regard to SMA aneurysms, as discussed above.

Lastly, we might confirm EVT as the proper current treatment, both urgent and elective, of intestinal aneurysms having the advantages of lower morbidity and mortality compared to open surgery, the last should be reserved to refractory cases, according to the most recent reports [2,53,71,72].

Evident limitations of this study include the small sample size of case series for these very rare conditions, the shortage of data reported, and the noncomparative uncontrolled nature of the available studies.

## 6. Conclusions

In conclusion, a high clinical suspicion for these uncommon VAAs, particularly when sources of gastrointestinal bleeding are not clearly identified, and a low threshold for treatment should be adopted, in relation to their high potential for rupture, no matter the size.

In addition, as clearly presented in our case, intestinal pseudoaneurysms should be listed among rare complications of IBD, especially CD, whose peculiar pathophysiology of transmural inflammation might extend to mesenteric vessels, thus hypothetically promoting the development of pseudoaneurysms.

Future research efforts could focus on establishing rigorous registries documenting the main aspects of all splanchnic aneurysms, such as site, size, underlying pathophysiology, clinical presentation, treatment choices, and outcomes.

## Figures and Tables

**Figure 1 medicina-58-01344-f001:**
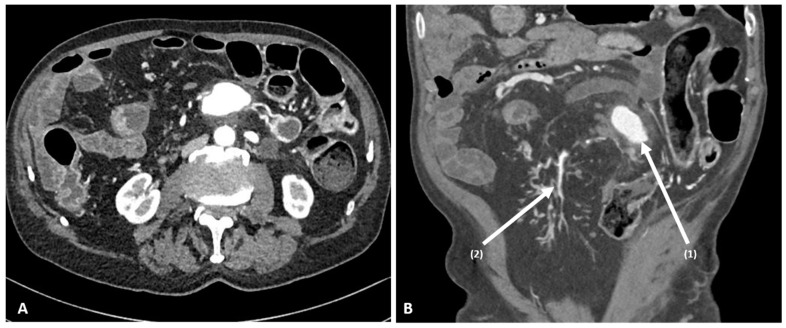
Computed tomography enterography (CTE) showing the jejunal artery (JA) pseudoaneurysm measuring 53 × 47 × 25 mm. (**A**) Axial image demonstrating the inflow and outflow arteries. (**B**) Coronal image displaying the aneurysm inside the small-bowel mesentery (arrow 1) and the relationship with the superior mesenteric artery (SMA) (arrow 2).

**Figure 2 medicina-58-01344-f002:**
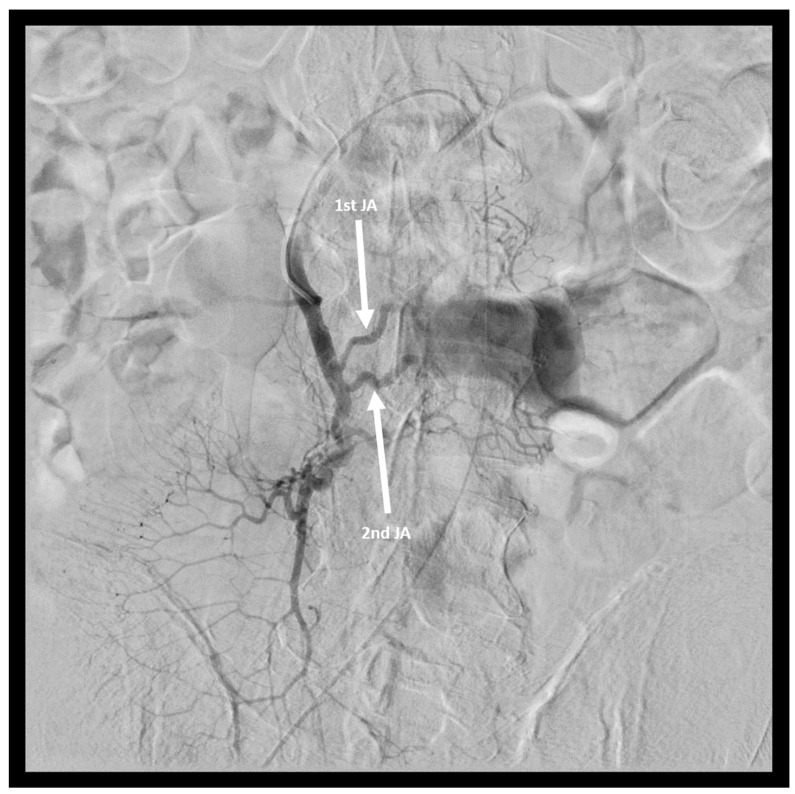
Angiography showing the pseudoaneurysm arising at the level of the 1st and 2nd JA (arrows).

**Figure 3 medicina-58-01344-f003:**
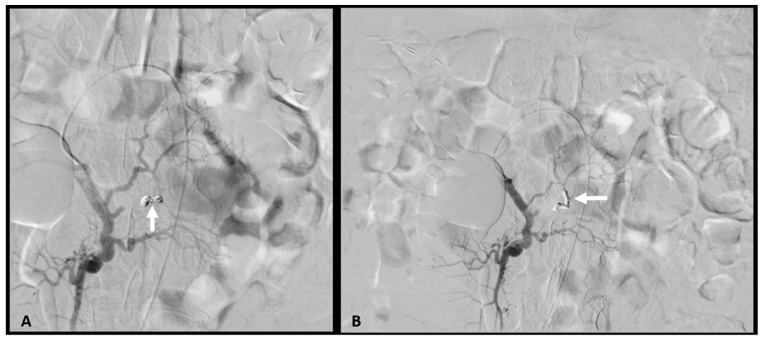
Transcatheter arterial embolization with coils of the pseudoaneurysm. (**A**) Angiogram after the first procedure showing complete occlusion of the 2nd JA (arrow). (**B**) Angiogram after the second procedure demonstrating complete occlusion of the distal branch of the 1st JA (arrow), with residual filling of the pseudoaneurysm trough the proximal branch.

**Figure 4 medicina-58-01344-f004:**
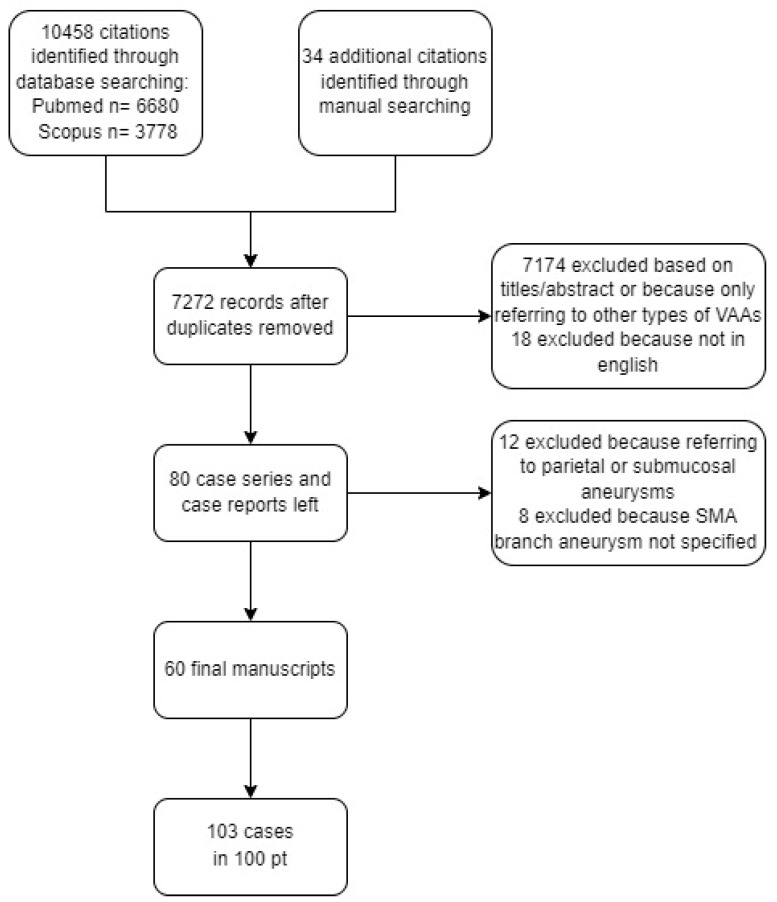
The flow chart outlining the strategy identifying eligible case studies.

**Table 1 medicina-58-01344-t001:** Case series of JA and IA aneurysms including at least 2 patients.

Author	Year	Total pt n.	JAAs/ IAAs pt n.	Age/ Sex	Site/ n	TRUE	Cause	Size (mm)	Rupture/ Pt n.	Main symptoms	MH/ HPT	Treatment	Outcome
Reuter [16]	1968	4	4	39/F	J		CONG	6	no		-	aneurysmectomy	uneventful
				72/F	I	yes	ATS	3.5	no	asymptomatic	-	-	-
				57/F	I		ATS	11	no		-	-	-
				76/M	I		ATS	4	yes	hypotension	MH	RHC	death (MI)
Hoehn [17]	1968	2	2	60/F	J	n.r.	ID	n.r.	yes	shock	MH	e-lap	uneventful
				50/F	J	yes	CONG	12	no	chronic pain	-	aneurysmectomy	uneventful
Stanley [4]	1970	45	3	71/M	J							-	-
				68/M	I	yes	ATS	n.r.	no	asymptomatic	-	-	-
				74/M	I							aneurysmectomy	uneventful
Tessier [36]	2002	12	5	69/F	J	n.r.	n.r.	30	yes	acute pain	no	ligation	uneventful
				48/F	J	n.r.	n.r.	15	yes	acute pain	no	ligation	uneventful
				79/M	J	n.r.	n.r.	10	no	asymptomatic	-	-	-
				56/M	J/1; I/1	no	PAN	n.r.	yes	acute pain	no	RHC	uneventful
				72/M	I	n.r.	n.r.	n.r.	no	asymptomatic	-	-	-
Roberts [53]	2015	48	3	n.r.	n.r.								death (MI)
				26/M	I	n.r.	n.r.	n.r.	no	GI bleeding	-	embolization	uneventful
				n.r.	n.r.								uneventful
Corey [6]	2016	250	3	n.r./M		yes	ATS	8				-	-
				n.r/M	J	no	VASC	30	no	asymptomatic	-	aneurysmectomy	uneventful
				n.r./F		n.r.	n.r.	30				CE	uneventful
Shimohira [67]	2021	45	4	n.r.	n.r.	yes	SAM	n.r.	yes/2	n.r.	n.r.	CE	uneventful
									no/2	asymptomatic	-	-	-
Pitcher [70]	2022	131	24	n.r.	J	n.r.	n.r.	n.r.	n.r.	n.r.	n.r.	n.r.	n.r.

JA, Jejunal Artery; IA, Ileal Artery; JAAs, Jejunal Artery Aneurysms; IAAs, Ileal Artery Aneurysms; MH, Mesenteric Hematoma; HPT, Hemoperitoneum; J, Jejunal; I, Ileal; CONG, Congenital; ATS, Atherosclerosis; RHC, Right Hemicolectomy; MI, Myocardial Infarction; n.r., not reported; ID, Idiopathic; e-lap, exploratory laparotomy; PAN, Polyarteritis Nodosa; GI, Gastro-Intestinal; VASC, Vasculitis; CE, Coil Embolization; SAM, Segmental Arterial Mediolysis.

**Table 2 medicina-58-01344-t002:** Single case reports of ruptured JA and IA aneurysms.

Author	Year	Age/ Sex	Site/ n	True	Cause	Size (mm)	Clinical Picture	Treatment	Outcome
Rathmell [13]	1951	29/F	J	yes	CONG	5	GI bleeding	e-lap	death
Horton [14]	1959	31/M	I	no	IE	50	acute pain, fever	SBR	uneventful
Hug [15]	1961	22/F	J	yes	CONG	5	GI bleeding, acute pain, hypotension	SBR	uneventful
Gueco [18]	1969	20/F	J	yes	CONG	8	GI bleeding	SBR	uneventful
Han [19]	1976	50/F	J	no	PAN	n.r.	GI bleeding, acute pain, shock, ME	RHC	death
Skudder [22]	1984	61/M	J	n.r.	n.r.	15	acute pain, shock, ME, HPT	ligation	death
Wilson [23]	1984	65/M	J	no	IE	15	GI bleeding	SBR	uneventful
Bleichrodt * [24]	1984	16/M	J	no	TR	n.r.	GI bleeding	ligation	uneventful
Diettrich [25]	1988	28/M	J	yes	CONG	6	GI bleeding shock	SBR	uneventful
Ishii [28]	1996	76/F	J/2	no	AP	10 both	shock, HPT	CE	uneventful
Rokke [30]	1997	73/M	J	no	abscess in PD	5	GI bleeding	CE	uneventful
Weinstock [31]	1999	57/M	J	n.r.	n.r.	5	SBO, ME	GJ	uneventful
Carr [33]	2001	n.r.	I	yes	ATS	n.r.	GI bleeding	ligation	n.r.
Oran [34]	2001	41/F	J	no	GI TB	5	GI bleeding	CE	uneventful
Ueda [35]	2001	74/M	I	no	MPA	n.r.	GI bleeding	CE	death
Kahn [41]	2006	12/F	J	no	GI TB	n.r.	GI bleeding	CE	uneventful
Bavunoglu [43]	2006	16/M	J	no	GI TB	n.r.	GI bleeding, acute pain, hypotension	GE	death
Asano [46]	2008	66/M	J	n.r.	n.r.	10	shock, HPT	aneurysmectomy	death
Garwood [48]	2009	54/F	J	yes	ATS	n.r.	shock, HPT, ME	aneurysmectomy	uneventful
Yamasaki [49]	2009	51/M	I	no	CSS	n.r.	GI bleeding	SBR	uneventful
Costa [52]	2013	76/M	J	yes	SAM	52	acute pain, hypotension, HPT	aneurysmectomy followed by SBR	uneventful
Wu [54]	2015	35/M	J	no	BD	4.8	acute pain, hypotension, HPT	SBR	uneventful
Ray [57]	2015	59/F	J	n.r.	n.r.	23	acute pain, hypotension, HPT	aneurysmectomy	uneventful
Arer [58]	2016	68/M	J	no	ACD	n.r.	GI bleeding	CE	uneventful
Kimura [63]	2018	47/F	I	no	stapler anastomosis	n.r.	acute pain	CE	uneventful
Murakawa [66]	2021	21/F	J	no	AAV	n.r.	acute pain, GI bleeding	embolization	uneventful

* The Author described two JA aneurysms in the same patient, of whom one ruptured (listed in Table 2) and the second non-ruptured (reported in Table 3). JA, Jejunal Artery; IA, Ileal Artery; J, Jejunal; CONG, Congenital; GI, Gastro-Intestinal; e-lap, exploratory laparotomy; I, Ileal; IE, Infective Endocarditis; SBR, Small Bowel Resection; PAN, Polyarteritis Nodosa; n.r., not reported; ME, Mesenteric Hematoma; RHC, Right Hemicolectomy; HPT, Hemoperitoneum; TR, Trauma; AP, Acute Pancreatitis; CE, Coil Embolization; PD, Pancreatico-duodenectomy; SBO, Small Bowel Obstruction; GJ, Gastro-jejunostomy; ATS, Atherosclerosis; TB, Tubercolosis; MPA, Microscopic Polyangitiis; GE, Glue Embolization; CSS, Churg-Strauss Syndrome; SAM, Segmental Arterial Mediolysis; BD, Behcet Disease; ACD, Acute Colonic Diverticulitis; AAV, ANCA-Associated Vasculitis.

**Table 3 medicina-58-01344-t003:** Single case reports of non-ruptured JA and IA aneurysms.

Author	Year	Age/ Sex	Site	True	Cause	Size (mm)	Clinical Picture	Treatment	Outcome
Keehan [20]	1978	21/M	J	no	IE	50	acute pain	aneurysmectomy	uneventful
McNamara [21]	1980	72/F	J	yes	ATS	40	acute pain, vomiting	aneurysmectomy	uneventful
Bleichrodt * [24]	1984	16/M	J	no	TR	n.r.	n.r.	ligation	uneventful
Ku [26]	1990	50/M	J	no	CP	12	chronic pain	PE	uneventful
Lindberg [27]	1992	44/M	J	no	IE	5	asymptomatic	-	-
Kubota [29]	1997	32/M	J	yes	IM	n.r.	asymptomatic	-	-
Dongola [32]	2000	39/M	J	no	primary APS	n.r.	GI bleeding, acute pain, shock	RPH evacuation	death
Gabelmann [37]	2002	78/M	J	no	IE	15	asymptomatic	CE	uneventful
Chiu [38]	2002	60/M	I	yes	ATS	30	acute pain	aneurysmectomy	uneventful
Morra [39]	2002	70/F	J	n.r.	n.r.	50	acute pain	aneurysmectomy	uneventful
Lorelli [40]	2003	58/F	J	no	CP	20	chronic pain	CE	uneventful
Shimohira [42]	2006	71/M	J	yes	ATS	10	asymptomatic	CE	uneventful
Sohn [44]	2007	73/F	J	yes	ATS	45	asymptomatic	aneurysmectomy+SVIG	uneventful
Yan [45]	2007	53/M	J	no	bacteriemia after ESWL	50	acute pain	SBR	uneventful
Turkbey [47]	2008	85/M	J	yes	ATS	5	GI bleeding	CE	uneventful
Kurdal [50]	2010	62/F	J	yes	ATS	45	acute pain	aneurysmectomy+SVIG	uneventful
Rossi [51]	2013	76/M	J	yes	ATS	12	asymptomatic	PE	uneventful
Lo [55]	2015	57/M	J	n.r.	n.r.	9	asymptomatic	CE	uneventful
Breguet [56]	2015	34/F	J	no	CP	17	chronic pain	CE	uneventful
Guirgis [59]	2017	86/M	J	yes	ATS	19	chronic pain	CE	uneventful
Kaihara [60]	2018	34/M	J	n.r.	n.r.	35	chronic pain	aneurysmectomy+SVIG	uneventful
Toya [61]	2018	59/M	J	no	heterotopic pancreas	17	asymptomatic	aneurysmectomy+SVIG	uneventful
Minaya-Bravo [62]	2018	49/F	J	yes	CONG	50	asymptomatic	aneurysmectomy	uneventful
Chen [64]	2021	56/F	J	no	AVM	18	GI bleeding	SBR	uneventful
Gogeneata [65]	2021	54/M	J	no	IE	39	acute pain, fever	aneurysmectomy	uneventful
Anwar [68]	2021	43/F	J	no	CP	n.r.	asymptomatic	-	-
Yadav [69]	2021	55/M	J	no	IgG4-vasculopathy	n.r.	asymptomatic	-	-

* The Author described two JA aneurysms in the same patient, of whom one ruptured (listed in Table 2) and the second non-ruptured (reported in Table 3). JA, Jejunal Artery; IA, Ileal Artery; J, Jejunal; IE, Infective Endocarditis; ATS, Atherosclerosis; TR, Trauma; n.r., not reported; CP, Chronic Pancreatitis; PE, Plug Embolization; IM, Idiopathic Medionecrosis; APS, Anti-phospholipid Syndrome; GI, Gastrointestinal; RPH, Retroperitoneal Hematoma; CE, Coil Embolization; I, Ileal; SVIG, Saphenous Vein Graft Interposition; ESWL, Extracorporeal Shock Wave Lithotripsy; SBR, Small Bowel Resection; CONG, Congenital; AVM, Arterio-Venous Malformation.

**Table 4 medicina-58-01344-t004:** Concomitant aneurysms identified in our literature review.

Author	Pt n.	Other VAAs	Site	Other Non VAAs	Site	Rupture/ Site	Treatment/ Site	Outcome
Hoehn [17]	1	yes	SA	no	-	no	no	uneventful
	2	no	-	yes	intracranial	no	no	uneventful
Stanley [4]	1	no	-	yes	AA	no	no	uneventful
	2	no	-	yes	AA	no	no	uneventful
Han [19]	1	yes	SA, HA, CA	no	-	yes/ SA, CA	splenectomy, RHC	death
Ku [26]	1	yes	GDA	no	-	no	no	uneventful
Lindberg [27]	1	yes	SMA	no	-	no	no	uneventful
Kubota [29]	1	yes	HA, SA, SMA, CA	yes	CIA	no	no	uneventful
Dongola [32]	1	yes	PDA CA	no	-	yes/PDA	RPH evacuation	death
Oran [34]	1	yes	jejunal vasa recta	no	-	no	no	uneventful
Tessier [36]	1	yes	RA, CTA	yes	thoracic,	no	no	uneventful
			SMA, IMA, CA		intracranial			
	2	no	-	yes	AA	no	no	uneventful
Rossi [51]	1	no	-	yes	AA	no	no	uneventful
Wu [54]	1	yes	SMA	no	-	no	no	uneventful
Anwar [68]	1	yes	CTA, PDA, GDA	no	-	no	CE/ PDA, GDA	uneventful
Yadav [69]	1	yes	HA, PDA, CTA	yes	coronary, intercostal	no	no	uneventful

VAAs, Visceral Artery Aneurysms; SA, Splenic Artery; AA, Abdominal Aorta; HA, Hepatic Artery; CA, Colic Artery; RHC, Right Hemicolectomy; GDA, Gastro-duodenal Artery; SMA, Superior Mesenteric Artery; CIA, Common Iliac Artery; PDA, Pancreatico-duodenal Artery; RPH, Retroperitoneal Hematoma; RA, Renal Artery; CTA, Celiac Trunk Artery; IMA, Inferior Mesenteric Artery; CE, Coil Embolization.

**Table 5 medicina-58-01344-t005:** Mortality and technical success rate after EVT of JA and IA aneurysms.

	Rupture Group (*n* = 33 pt)	No-rupture Group (*n* = 43 pt)
*Mortality, n (%)*	7 (21.2)	2 (4.6)
Aneurysm-related mortality	4 (57.1)	1 (50)
*Endovascular technical success rate, n (%)*	11 (100)	12 (92.3)

EVT, Endovascular Treatment; JA, Jejunal Artery; IA, Ileal Artery.

## Data Availability

All the data are deposited in a database stored at the Ospedali Riuniti Marche Nord.

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
