# Peer review of "Crohn’s Disease and Jejunal Artery Aneurysms: A Report of the First Case and a Review of the Literature"

_medicina, 2022, doi:10.3390/medicina58101344_

Round 1

Reviewer 1 Report

Very good work. The authors provide helpful informationns. The prompt treatment accomplished granted a successful outcome. JA and IA aneurysms should be included among local complications of IBD. Considering their high potential for rupture, regardless of size, a low threshold for endovascular or surgical treatment should be applied.

Reviewer 2 Report

Thank you for the opportunity to review this excellent case report and review of the literature concerning the demographics and surgical management of jejunal and ileal arterial aneurysms (JA and IA). JA and IA are extremely rare examples of visceral arterial aneurysms (3% of VAAs), hence this manuscript addresses a surgical issue that has not previously been well covered in the surgical literature. Current guidelines from the Society for Vascular Surgery are based on only 15 cases of JA/IA, while this manuscript reports data from 103 cases.

The authors present a case report of an exacerbation of Crohn's disease within the small bowel, where a large (diameter 25 mm) pseudoaneurysm of branches of the Jejunal artery was likely incidentally detected. Endovascular ablation was undertaken, but due to residual filling, open surgical resection of the mesentery containing the aneurysm and the associated segment of small bowel was subsequently undertaken, with success. The authors then undertook an exhaustive review of the literature, resulting in the analysis of 103 cases of jejunal or ileal aneurysm reported over the last 70 years. 

The authors draw a number of very relevant conclusions in relation to surgical management of these aneurysms, including some useful demographics and disease associations that should assist in diagnosis. In particular, they confirm that pseudoaneurysms are less stable and therefore elective repairs are required. The authors discuss these findings in relation to the current guidelines from the Society for Vascular Surgery. The authors note that endovascular repairs are generally successful, with open repair being required in only a small proportion of cases. Interestingly, ruptured aneurysms are generally smaller than non-ruptured aneurysms. The authors appropriately identify limitations to their study, mainly related to the small sample size and the paucity of data in some of the previous papers they identified.

I have only a few minor comments that I suggest the authors may wish to address:

1. The authors report that histopathological examination of their aneurysm was undertaken, but do not report the microscopic findings of changes within the arterial wall. For example, was arteritis present, and if so what were the particular features? A useful related reference that the authors may wish to include is:

Sy A, Khalidi N, Dehghan N, Barra L, Carette S, Cuthbertson D, Hoffman GS, Koening CL, Langford CA, McAlear C, Moreland L, Monach PA, Seo P, Specks U, Sreih A, Ytterberg SR, Van Assche G, Merkel PA, Pagnoux C; Vasculitis Clinical Research Consortium (VCRC); Canadian Vasculitis Network (CanVasc). Vasculitis in patients with inflammatory bowel diseases: A study of 32 patients and systematic review of the literature. Semin Arthritis Rheum. 2016 Feb;45(4):475-82. doi: 10.1016/j.semarthrit.2015.07.006. Epub 2015 Jul 26. PMID: 26315859; PMCID: PMC4982464. 

This issue is of relevance because the authors suggest an association between Crohn's disease and the formation of the VAA they reported. While this is likely, it is also possible that the aneurysm is unrelated to Crohn's (which the authors acknowledge), for example, it may simply be atherosclerotic.

2. Grammatical errors and minor changes:

(i) Abstract line 16: One third of patients(32.4%) WERE asymptomatic...

(ii) Page 2, para 2, line 3: whereas those INVOLVING the gastric-gastroepiploic.....

(iii) Figure 4: The text in two of the boxes seems to have spilt out of the box and hence is partially illegible.

(iv) Page 12, para 1, line 1: intestinal aneurysms have been constantly reported TO BE the most uncommon....

(v) Page 12, para 4, line 1: Indeed, though PREVIOUSLY BEING generally described AS asymptomatic [1,72,73], our review....

(vi) Page 12, para 8, lines 1: Nevertheless, we could NOT find in THE literature any associations reported between IBD.....

(vii) Page 12, para 8, line 2: While the authors are correct that there has not been any association (that they or I could find) between visceral aneurysm and IBD, there is a relevant paper that describes an association between vasculitis and IBD (already mentioned above):

Sy A, Khalidi N, Dehghan N, Barra L, Carette S, Cuthbertson D, Hoffman GS, Koening CL, Langford CA, McAlear C, Moreland L, Monach PA, Seo P, Specks U, Sreih A, Ytterberg SR, Van Assche G, Merkel PA, Pagnoux C; Vasculitis Clinical Research Consortium (VCRC); Canadian Vasculitis Network (CanVasc). Vasculitis in patients with inflammatory bowel diseases: A study of 32 patients and systematic review of the literature. Semin Arthritis Rheum. 2016 Feb;45(4):475-82. doi: 10.1016/j.semarthrit.2015.07.006. Epub 2015 Jul 26. PMID: 26315859; PMCID: PMC4982464.

(viii) Page 14, para 1, line 1: Particularly, in the urgent setting of rupture and gastrointestinal bleeding, EVT has BEEN demonstrated TO BE very useful....
